



Simultaneous Measurement of Urban and Rural Single Particles in Beijing, Part I:
Chemical Composition and Mixing State
Yang Chen[1], Jing Cai[2], Zhichao Wang[1], Chao Peng[1], Xiaojiang Yao[1], Mi Tian[1], Yiqun
Han[2], Guangming Shi[1,3], Zongbo Shi[4,5], Yue Liu[2], Xi Yang[2], Mei Zheng[2*], Tong Zhu[2],
Kebin He[6], Qiang Zhang[7], and Fumo Yang[3,1*]
[1]Center for the Atmospheric Environment Research, Chongqing Institute of Green and
Intelligent Technology, Chinese Academy of Sciences, Chongqing 400714, China
[2]SKL-ESPC and BIC-ESAT, College of Environmental Sciences and Engineering, Peking
University, Beijing 100871, China
[3]College of Architecture and Environment, Sichuan University, Chengdu 610065, China
[4] School of Geography Earth and Environmental Sciences, University of Birmingham,
Birmingham B15 2TT, UK
[5]Institute of Surface-Earth System Science, Tianjin University, Tianjin 300072, China
[6] School of Environment, Tsinghua University, Beijing 100084, China
[7]Department of Earth System Science, Tsinghua University, Beijing, China
Corresponding    to    Fumo    Yang    (fmyang@scu.edu.cn)    and    Mei    Zheng
(mzheng@pku.edu.cn)
Keywords: urban; regional; single particle; transport; mixing state





**Abstract**
Two single particle aerosol mass spectrometers (SPAMS) were deployed simultaneously
at an urban and a rural site in Beijing during an intensive field campaign from 1$^{st}$ to 29$^{th}$
Nov 2016 to investigate the source and process of airborne particles in Beijing. In the first
part of this research, we report the single-particle chemical composition, mixing state, and
evolution at both sites. 96% and 98% of collected particles were carbonaceous at the urban
and rural sites, respectively. Five particle categories, including elemental carbon (EC),
organic carbon (OC), internal-mixed EC and OC (ECOC), potassium-rich (K-rich), and
Metals were observed at both sites. The categories were partitioned into particle types
depending on different atmospheric processing stages. Seventeen particle types were
shared at both sites. In the urban area, nitrate-containing particle types, such as EC-Nit and
ECOC-Nit, were enriched, especially at night; sulfate-containing particles were transported
when wind speed was high; ECOC-Nit-Sul were mostly local-aged. In sum, these
processed particles took up to 85.3% in the urban areas. In the rural area, regional particles
were abundant, but freshly emitted ECOC and OC had distinct patterns that were
pronounced at cooking and heating time. Biomass burning, traffic, and coal burning were
major sources of PM$_{2.5}$ in both rural and urban areas. Besides, the particles from the steel
industry located in the south were also identified. In summary, the chemical composition
of urban and rural particle types was similar in Beijing; the urban particles were influenced
significantly by rural processing and transport. The work is useful to understand the
evolution of urban and rural particles in Beijing during winter.





**1. Introduction**
China has experienced severe haze events caused by extremely high concentrations of fine
particulate matter (PM$_{2.5}$) since January 2013. In the worst cases, an area of 2.0 million
km$^2$ and a population of 800 million were affected (Huang et al., 2014). In the Beijing-
Tianjin-Hebei (BTH) area, extreme haze events frequently occur during winter, with PM$_{2.5}$
mass reaching rapidly up to 200 µg m$^{-3}$ and sustaining such levels for hours (Guo et al.,

46  2014).

Over the last two decades, comprehensive studies have been conducted on urban PM in
Beijing. He et al. (2001) reported the first characterization of PM$_{2.5}$. Since then, numerous
studies have been published on characterization (Huang et al., 2010), sources (Guo et al.,
2012; Sun et al., 2014), and processing of PM (Sun et al., 2013). The mechanism of rapid-
boosting PM$_{2.5}$ in Beijing, including new particle formation and growth (Guo et al., 2014),
regional transport (Li et al., 2015), and both (Du et al., 2017; Sun et al., 2014), have been
proposed. However, discrepancies remain among these studies.
Single particle mass spectrometers (SPMS) have been used to investigate the size-resolved
chemical composition and mixing state of atmospheric particles (Gard et al., 1997; Pratt
and Prather, 2012). More recently, single particle aerosol mass spectrometers (SPAMS)
have been used in Chinese megacities such as Beijing (Li et al., 2014), Shanghai (Tao et
al., 2011), Guangzhou (Bi et al., 2011), Xi'an (Chen et al., 2016), Nanjing (Wang et al.,
2015), and Chongqing (Chen et al., 2017).





In Beijing, particle types, such as carbonaceous, metal, dust, K-rich, and others during
spring and fall, were reported (Liu et al., 2016b; Li et al., 2014). Besides, lead-containing
particles have also been investigated in recent studies (Ma et al., 2016; Cai et al., 2017).
However, due to the insufficient consideration of mixing state of nitrate, sulfate, and
organics, these studies paid limited attention to the atmospheric particulate processing.
This study is a part of the APHH-Beijing (Atmospheric Pollution and Human Health in a
Chinese Megacity of Beijing) intensive field campaign during winter 2016 (Shi et al., 2019).
Two SPAMSs were deployed simultaneously at Peking University (PKU) and Pinggu (PG)
in order to observe both urban and rural particles in the Beijing region. The aims of the
study are 1) to characterize the single-particle chemical composition and mixing state; 2)
to investigate particulate evolution at both sites during haze events. These two objectives
are presented in two parts. In Part I, particle types and their atmospheric processing (e.g.,
origination, source, and diurnal profiles) at both sites are reported; in Part II, the detailed
analysis of haze events, effects of heating activities, and evidence of regional transport are
addressed.
**2. Methodology**
**2.1 Sampling sites**
The campaigns were performed simultaneously at PKU (116.32ºE, 39.99ºN) and PG
(117.05ºE, 40.17ºN) from 11/01/2016 to 11/29/2016. A Description of the PKU site is
available in the literature (Huang et al., 2006). Briefly, the site is located on the rooftop (15
m above the ground) on the PKU campus which is surrounded by residential and





commercial blocks. Trace gases (Thermo Inc. series), meteorological parameters (Vaisala
Inc.), and $PM_{2.5}$ (TEOM 1430) were recorded during the observation.
The PG site (117.053ºE, 40.173ºN) is 3 km from the PG center. The site is located in the
northeast of the PKU site with a distance of 70 km. The PG site also acts as a host of the
AIRLESS (Effects of AIR pollution on the cardiopulmonary disease in urban and peri-
urban residents in Beijing) Project. The meteorological data is acquired from the local
meteorological office. The PG village is surrounded by orchards and farmland with no
main road nearby on a scale of 3 km. Coal and biomass are used for domestic heating and
cooking in the nearby villages.
**2.2 Instrumentation and data analysis**
Two SPAMSs (Model 0515, Hexin Inc., Guangzhou, China) were deployed at both PKU
and PG. A technical description of SPAMS is available in (Li et al., 2011). Briefly, a
SPAMS has three functional parts: sampling, sizing, and mass spectrometry. In the
sampling part, particles within a 0.1–2.0 µm size range pass efficiently through an
aerodynamic lens. In the sizing unit, the aerodynamic diameter ($D_{va}$) is calculated using
the time-of-flight of particles. The particles are then decomposed and ionized into ions one-
by-one using a 266 nm laser. A bipolar time-of-flight mass spectrometer measures the ions
and generates the positive and negative mass spectra of each particle. The two instruments
were maintained and calibrated following the standard procedures before sampling (Chen
et al., 2017).





A neural network algorithm based on adaptive resonance theory (ART-2a) was used to
resolve particle types from both datasets (Song et al., 1999). The parameters used were: a
vigilance factor of 0.70, a learning rate of 0.05, and 20 iterations. This procedure generated
771 and 792 particle groups. Then, the groups were combined into particle types based on
similar mass spectra, temporal trends, and size distributions (Dallosto and Harrison, 2006).
During combining, relative areas of nitrate and sulfate were used to distinguish the stages
of processing, assuming that more sulfate and nitrate can be measured if a particle is more
processed during its lifetime. Thus, particles with relative peak areas of sulfate and nitrate
larger than 0.1 were marked with nitrate (-Nit), sulfate (-Sul), respectively, or both. Finally,
the strategy resulted in 20 and 19  particle types at PKU and PG respectively. Among them,
17 types appeared at both sites, and each type has identical mass spectra ($R^2 > 0.80$) between
each other.
**3. Results**
A total of 4,499,606 and 4,063,522 particles were collected at PKU and PG sites,
respectively. The size distributions peaked at 0.48 µm and 0.52 µm (Figure 1). The smaller
size distribution was due to a more substantial fraction of freshly-emitted particles at PG,
as described in Table 1. Seventeen particle types ($R^2 > 0.80$, mass spectra) were observed
both at PKU and PG (Table 1). These particle types were labeled with the suffixes "_PKU"
or "_PG" to indicate their locations. The term "particle category" stands for a group of
particle types with variable stages of processing.





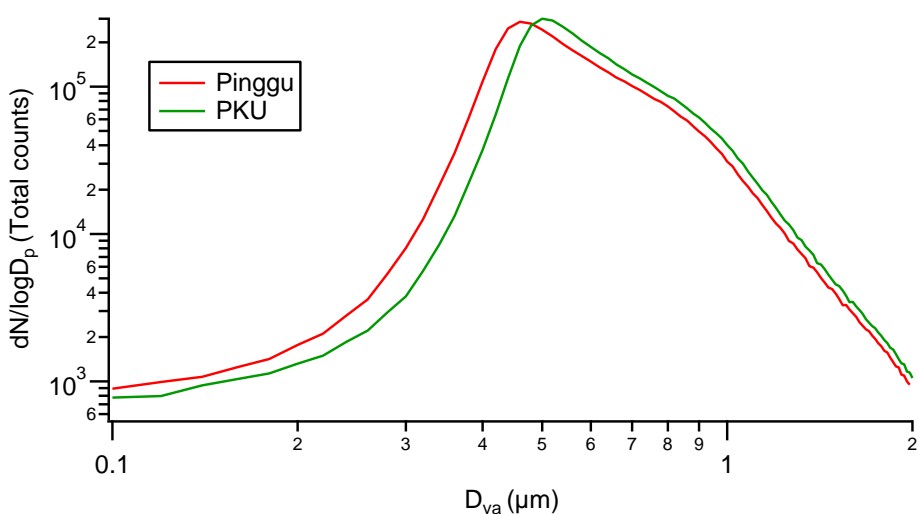


Figure 1. The size distribution of SPAMS particles at PKU and PG sites.



Table 1. SPAMS particle types identified at PKU and PG sites.

| At both sites | Particle type | PKU Number | PKU Percentage | PG Number | PG Percentage |
|---|---|---|---|---|---|
| EC | EC-Nit | 313574 | 7.0 | 79082 | 2.0 |
|  | EC-Nit-Sul | 473908 | 10.5 | 140107 | 3.5 |
|  | EC-Sul | 30365 | 0.7 | 4096 | 0.1 |
|  | ECOC-Nit-Sul | 539533 | 12.0 | 755279 | 18.6 |
|  | ECOC-Sul | 572548 | 12.7 | 397367 | 9.8 |
| K-rich | K-rich | 322731 | 7.2 | 259287 | 6.4 |
|  | K-Nit | 359281 | 8.0 | 334547 | 8.2 |
|  | K-Nit-Sul | 717280 | 16.0 | 76954 | 1.9 |
|  | K-Sul | 26301 | 0.6 | 183571 | 4.5 |
| NaK | NaK | 16680 | 0.4 | 74943 | 1.8 |
|  | NaK-Nit | 289259 | 6.4 | 69760 | 1.7 |
|  | NaK-Nit-Sul | 114387 | 2.5 | 77555 | 1.9 |
|  | NaK-Sul | 7509 | 0.2 | 16578 | 0.4 |
| OC | OC-Nit-Sul | 334870 | 7.4 | 865821 | 21.3 |
|  | OC-Sul | 40800 | 0.9 | 279322 | 6.9 |
|  | Ca-dust | 19869 | 0.4 | 3035 | 0.1 |
| Metal | Fe-rich | 137600 | 3.1 | 70920 | 1.8 |
|  | ECOC-Nit | 137470 | 3.1% |  |  |
|  | OC-Nit | 41159 | 0.9% |  |  |
|  | K-Amine-Nit-Sul | 4482 | 0.1% |  |  |
|  | ECOC |  |  | 239953 | 5.9% |
|  | OC |  |  | 135345 | 3.3% |

Note: Nit stands for nitrate, Sul for sulfate.

**3.1 Meteorological conditions and overview**

Temperature, relative humidity (RH), and wind speed at both sites during the sampling
period are summarized in Table 2. Their temporal trends are available in Part II. The
average temperature at PKU (urban, 5.7±2.3 °C) was higher than at PG (rural, 3.1±2.2 °C).
Correspondingly, relative humidity was higher at PG (67±32%) than at PKU (49±30%).
The wind was stronger at PKU (2.5± 1.8 ms$^{-1}$) than at PG (1.7± 0.9 ms$^{-1}$). As shown in





Figure 2, at PKU, wind speed peaked at noon (local time, UTC+8), while at PG, wind speed
reached its maxima at 15:00. Various wind speeds determined the different dispersion
patterns of pollutants near the surface. It should be noticed that wind speed up to 2 ms$^{-1}$
representing a scale of 172 km in diurnal transport. Therefore, at PKU, the wind could
bring the pollutants from Hebei province under stagnant air conditions.

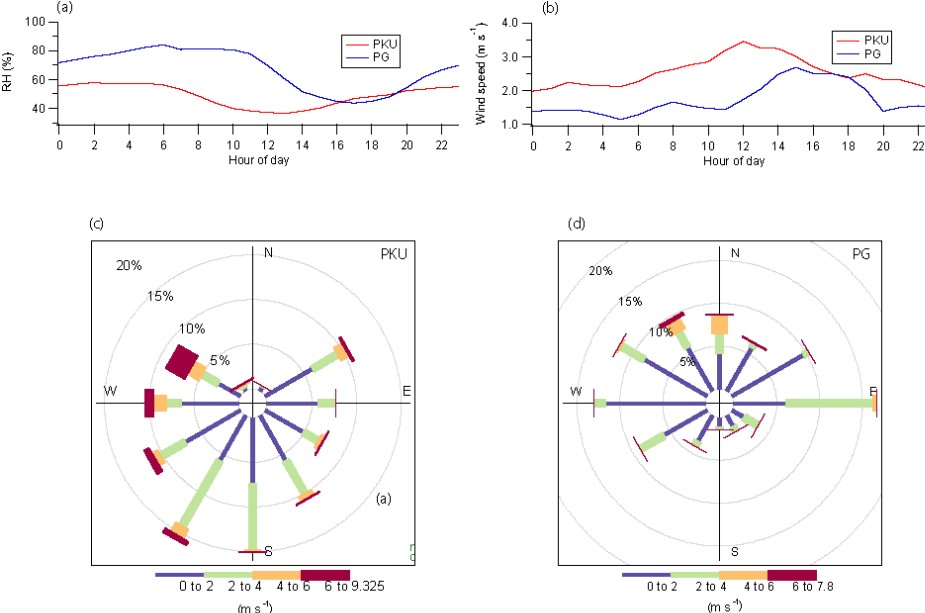


Figure 2. Diurnal plots of (a) RH and (b) wind speed, and rose plots of wind at (c)PKU and
(d) PG.
Table 2. Meteorological parameters at PKU and PG during the campaign.

|  | PKU | PG |
|---|---|---|
| Temperature (°C) | 5.7±2.3 | 3.1±2.2 |
| RH (%) | 49±30 | 67±32 |
| Wind speed (ms$^{-1}$) | 2.5± 1.8 | 1.7± 0.9 |




### 3.2 Common particle categories at both PKU and PG

### 3.2.1 Elemental carbon (EC)

As shown in Figure 3a, the elemental carbon (EC) particle category was represented by ions peaking at $m/z$ 12, 24, 36, 48, and 60 in positive mass spectra (Sodeman et al., 2005; Toner et al., 2008). EC is emitted from solid fuel combustion, traffic (Sodeman et al., 2005; Toner et al., 2008; Toner et al., 2006), and industrial activities (Healy et al., 2012). Due to the various ionic intensities of nitrate ($m/z$ -46 and -62) and sulfate ($m/z$ -80 and -97), the EC category has four types including EC-Nitrate (EC-Nit), EC-Sulfate (EC-Sul), and EC-Nit-Sul. Besides, the EC category was more abundant after the heating began rather than before (Part II), indicating that coal burning was one of the primary sources.

EC-Nit_PKU and EC-Nit_PG accounted for 7.0% and 2.0% in PKU and PG datasets, respectively. In the diurnal profiles of EC-Nit_PKU, there was an apparent early morning peak at 5:00 (UTC+8, local time), along with an evening peak (22:00). There was also an early morning $NO_x$ peak in the urban area of Beijing, providing sufficient precursors for secondary nitrate (Shi et al., 2019). Wang et al. (2018) validated the role of $N_2O_5$ uptake on the nitrate formation in PM. Therefore, the early morning peak of EC-Nit_PKU occurred due to the uptake of nitrate on the freshly emitted EC in the early morning (Sun et al., 2014). The evening peak could be due to the low temperature after the heating supply started. Diurnally, EC-Nit_PG exhibited an early morning peak (5:00) but no evening peak and mainly came from the southeast.

EC-Nit-Sul was more abundant at the rural site (18.6%) than the urban site (11.6%). EC-Nit-Sul_PKU (10.5%) had early morning (04:00), morning (7:00), and afternoon peaks





(around 16:00), while EC-Nit-Sul_PG (3.5%) had early morning (04:00), noon, and
afternoon peaks (17:00, Figure 3a). However, they showed relatively small diurnal
variations. For example, EC-Nit-Sul_PKU varied between 800 h$^{-1}$ and 1,000 count h$^{-1}$, and
EC-Nit-Sul_PG shifted between 200 count h$^{-1}$ and 250 count h$^{-1}$. Thus, the EC-Nit-Sul at
both sites was most likely acting as background and regional particles (Dall'Osto et al.,
2016). Additionally, EC-Nit-Sul_PKU mainly came from the surrounding area in the city
pollutant plume, while EC-Nit-Sul_PG mainly came from the southeast (Figure 3c).
EC-Sul was a minor type at both sites, accounting for 0.7% at PKU and 0.1% at PG. EC-
Sul was pronounced in the afternoon when the wind was strong at both sites. It was unlikely
for either EC-Sul_PKU or EC-Sul_PG to be local because their concentrations were
associated with high wind speed, as shown in Figure 3c. More specifically, EC-Sul_PKU
came from the southeast and northeast of Hebei Province when the wind speed exceeded
6 m s$^{-1}$. EC-Sul_PG could come from the west when the wind speed exceeded 2 m s$^{-1}$ and
the east when the wind speed exceeded 3 m s$^{-1}$, as coal-using industries are located in both
directions. Also, at both sites, the concentrations of SO$_2$ were elevated in the afternoon due
to transport, providing sufficient precursors for the formation of sulfate (Shi et al., 2019).

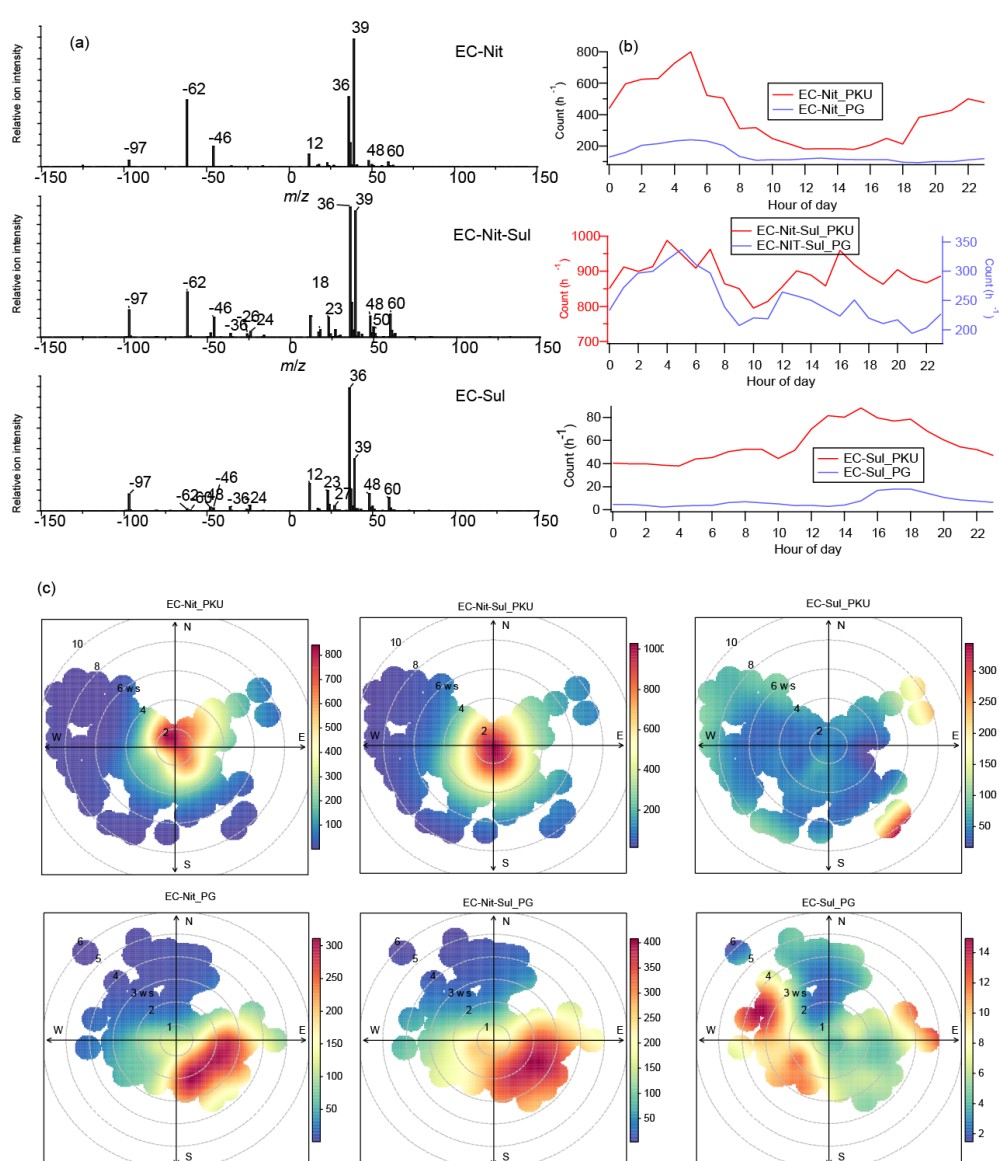

Figure 3. (a) average mass spectra of EC-Nit, EC-Nit-Sul, and EC-Sul at both sites; (b)
diurnal patterns of EC-Nit, EC-Nit-Sul, and EC-Sul at both sites; (c) polar plots of EC-Nit,
EC-Nit-Sul, and EC-Sul; the grey circles indicate wind speed (m s$^{-1}$).



### 3.2.2 Organic carbon (OC) category


The positive mass spectra of both OC-Nit and OC-Nit-Sul contained complicated organic
ions such as $C_2H_3^+$ ($m/z$ 27), $C_3H^+$ ($m/z$ 37), $C_3H_7^+/C_2H_3O^+/ CHNO^+$ ($m/z$ 43), $C_4H_2^+$ ($m/z$
50), aromatic hydrocarbons ($C_4H_3^+$, $C_5H_3^+$, and $C_6H_5^+$), and diethylamine (($C_2H_5$)$_2$NH$_2^+$,
$m/z$ 74), (($C_2H_5$)$_2$NCH$_2^+$ ($m/z$ 86)). The negative mass spectra contained $CN^-$ ($m/z$ −26), $Cl^-$
($m/z$ −35 and 37), $CNO^-$ ($m/z$ −42), nitrate ($m/z$ −46 and −62), and sulfate($m/z$ −97). The
presence of $CN^-$ and $CNO^-$ suggests the existence of organonitrogen species (Day et al.,
2010). Peak intensities of organic fragments are relatively high in the OC-Sul particles,
indicating that it was relatively fresh, while OC-Nit-Sul was more processed (Zhai et al.,
2015). The positive mass spectrum had similar ions of Coal Combustion OA (CCOA) with
significant signals of PAHs in AMS studies (Sun et al., 2013). OC-Sul showed different
spatial distributions with 0.9% at PKU and 6.9% at PG.
OC-Sul_PG had morning (8:00) and afternoon (16:00) peaks, while the diurnal profile of
OC-Sul_PKU showed a trend with an early morning (3:00), morning (10:00), and
afternoon peaks (16:00). The diurnal trends OC-Sul at both PKU and PG were consistent
with the heating pattern depending on the variation of local temperature. Moreover, OC-
Sul_PG increased after the heating supply began. Polar plots suggest that OC-Sul_PKU
came from surrounding southwest areas via transport, while OC-Sul_PG came from
villages to the east and west (Figure 4). These results suggest that OC-Sul_PG was emitted
from coal burning for residential heating in nearby areas.
OC-Nit-Sul accounted for 7.4 % and 21.3 % of all detected particles at PKU and PG,
respectively. OC-Nit-Sul_PKU had a diurnal peak at 7:00 in rush hours, suggesting that
OC-Nit-Sul could be formed due to the uptake of nitrate on OC-Sul. While OC-Nit-Sul_PG
had a diurnal peak at 8:00 due to traffic in nearby towns. As an aged particle type, OC-Nit-
Sul_PKU and OC-Nit-Sul_PG, also acting as a similar type of background types with
hourly counts remained low but elevated to high levels at night. Polar plots suggest that
OC-Nit-Sul_PKU mainly came from the surrounding areas, while OC-Nit-Sul_PG mainly
came from the south and east, where populous villages are located (Figure 4).

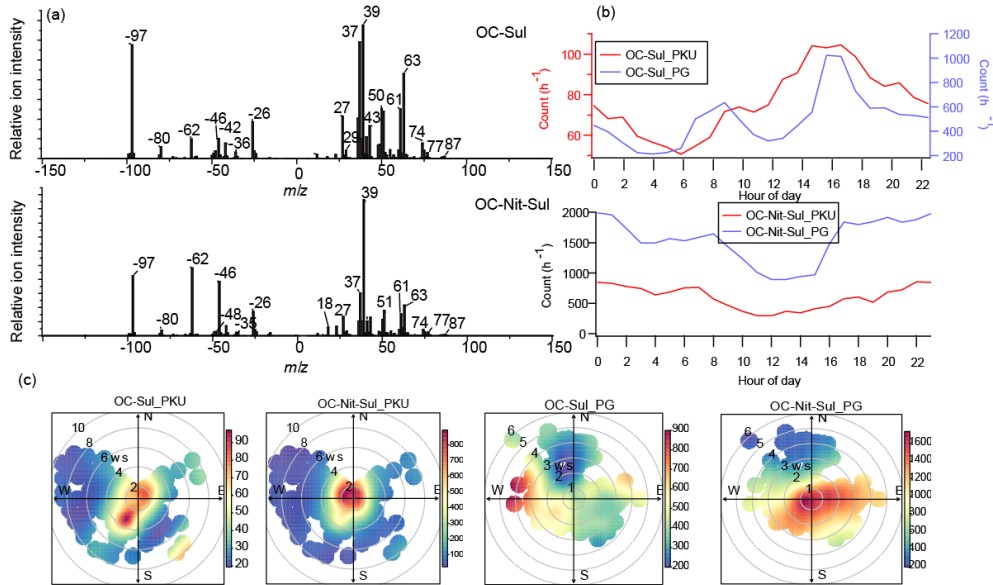


Figure 4. (a): average mass spectra of OC-Nit and OC-Nit-Sul observed at both sites; (b):
diurnal patterns of the hourly count of OC-Nit and OC-Nit-Sul at both sites; (c): polar plots
of OC-Sul and OC-Nit-Sul; the grey circles indicate wind speed (m s$^{-1}$).
**3.2.3 ECOC category**
As shown in Figure 5a, the ECOC category contained two major particle types: ECOC-
Nit-Sul and ECOC-Sul. The positive mass spectrum of ECOC-Nit-Sul contained $C_n^+$ (*m/z*



12, 24, 36…), $NH_4^+$ ($m/z$ 18), $C_2H_3^+$ ($m/z$ 27), $K^+$ ($m/z$ 39 and 41), $C_3H_7^+/C_2H_3O^+/$ $CHNO^+$
($m/z$ 43), $C_4H_2^+$ ($m/z$ 50), and $[(C_2H_5)_2NH_2]^+$ ($m/z$ 74); in the negative mass spectrum, ions
such as sulfate ($m/z$ -80 and -97), nitrate ($m/z$ -46 and -62), $C_n^-$, and $CN^-$ ($m/z$ -26) were
abundant. This mixture of EC and OC particle types was common in single particle studies.
ECOC could be local, and from incomplete combustion processes (Chen et al., 2017), or
regional transport, e.g., after aging (McGuire et al., 2011; Huang et al., 2013). The diurnal
profile of ECOC-Sul_PG showed early morning (1:00), morning (8:00), and afternoon
(17:00) peaks, which is consistent with local cooking and heating patterns. Also, heating
activities enhanced the fraction of ECOC-Sul_PG. ECOC-Sul_PKU did not show a clear
diurnal profile, suggesting that ECOC-Sul_PKU was mainly a background type. Similarly,
ECOC-Nit-Sul_PKU and ECOC-Nit-Sul_PG were also background types with less
obvious diurnal variations (Dall'Osto et al., 2016). Polar plots (Figure 5c) suggested that
both ECOC-Nit-Sul_PKU and ECOC-Sul_PKU had both local and regional sources. Wind
speed up to 4 m $s^{-1}$ could cause a transport with a distance of 346 km diurnally, indicating
that it was possible for the particles from Hebei province to arrive at the sampling place.



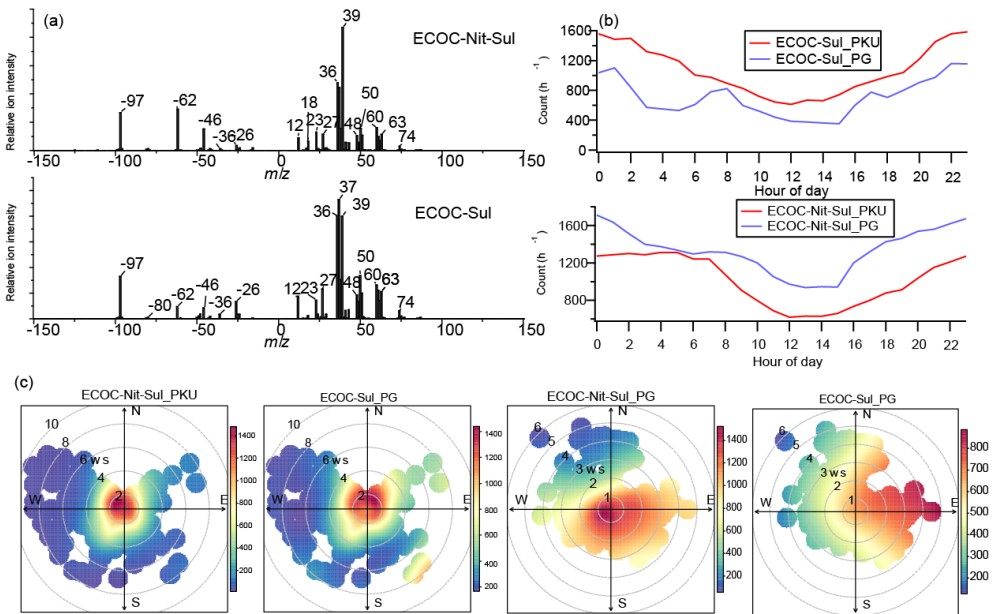


Figure 5. (a): average mass spectra of ECOC-Nit and ECOC-Nit-Sul observed at both sites;

(b): diurnal patterns of the hourly count of ECOC-Sul and ECOC-Nit-Sul at both sites; (c):

polar plots of ECOC-Sul and ECOC-Nit-Sul; the grey circles indicate wind speed ($ms^{-1}$).

**3.2.4 K-rich category**

Figure 6 shows a series of potassium-rich (K) particle types. K-rich contained $Na^+$ ($m/z$ 23),

$C_2H_3^+$ ($m/z$ 27), $C_n^+$, $C_3H^+$ ($m/z$ 37), $K^+$, aromatic hydrocarbons ($C_4H_3^+$, $C_5H_3^+$, and $C_6H_5^+$),

levoglucosan ($m/z$ -45, -59, and -71), sulfate, and nitrate. According to the ionic intensities

of sulfate and nitrate, the K-rich particle category had several branches such as K-rich, K-

Nit, K-Sul, and K-Nit-Sul. K-rich particles are commonly found in biomass burning

emissions (Silva et al., 1999; Pagels et al., 2013; Chen et al., 2017). $Cl^-$ was unabundant in

all K-rich particle types, suggesting that the K-rich particles had undergone aging during





atmospheric processing (Sullivan et al., 2007; Chen et al., 2016), but K-Nit, K-Nit-Sul, and
K-Sul were more processed.
All K-rich category particles showed different atmospheric evolution process at both PKU
and PG. K-rich_PKU illustrated a typical pattern that was at low levels in the daytime but
high levels at nighttime (22:00). As shown in Figure 6c, at an average wind speed of 3 m
$s^{-1}$, it took five hours for particles from a distance of 50 km to arrive at PKU. This is also
the reason why BB-related particles were abundant in urban Beijing where the household
BB is prohibited. The origination of K-rich_PKU was from nearby and southwest. K-
rich_PG, however, showed a pattern with cooking and heating activities, peaking at 7:00
and 17:00. The peak at 7:00 was due to the local emissions; the 17:00 could be transported
from a distance of 50 km at a wind speed of 3 m $s^{-1}$ from the east and west.
The secondary process contributed to the early morning peak (5:00) of K-Nit_PKU due to
the nighttime formation of nitrate via hydrolysis of $N_2O_5$ in the $NO_x$-rich urban areas (Wang
et al., 2017). In the day time, after the rush hours, the number concentration of K-Nit_PKU
increased again via the uptake of nitrate due to day time photoactivity. K-Nit_PKU mainly
originated from the local and southerly areas (Figure 6c).  Besides the early morning peak,
K-Nit_PG showed cooking and heating patterns that they were abundant when the
temperature was low in the early morning and afternoon. K-Nit_PG had wide originated
from both local and region via long-range transport.



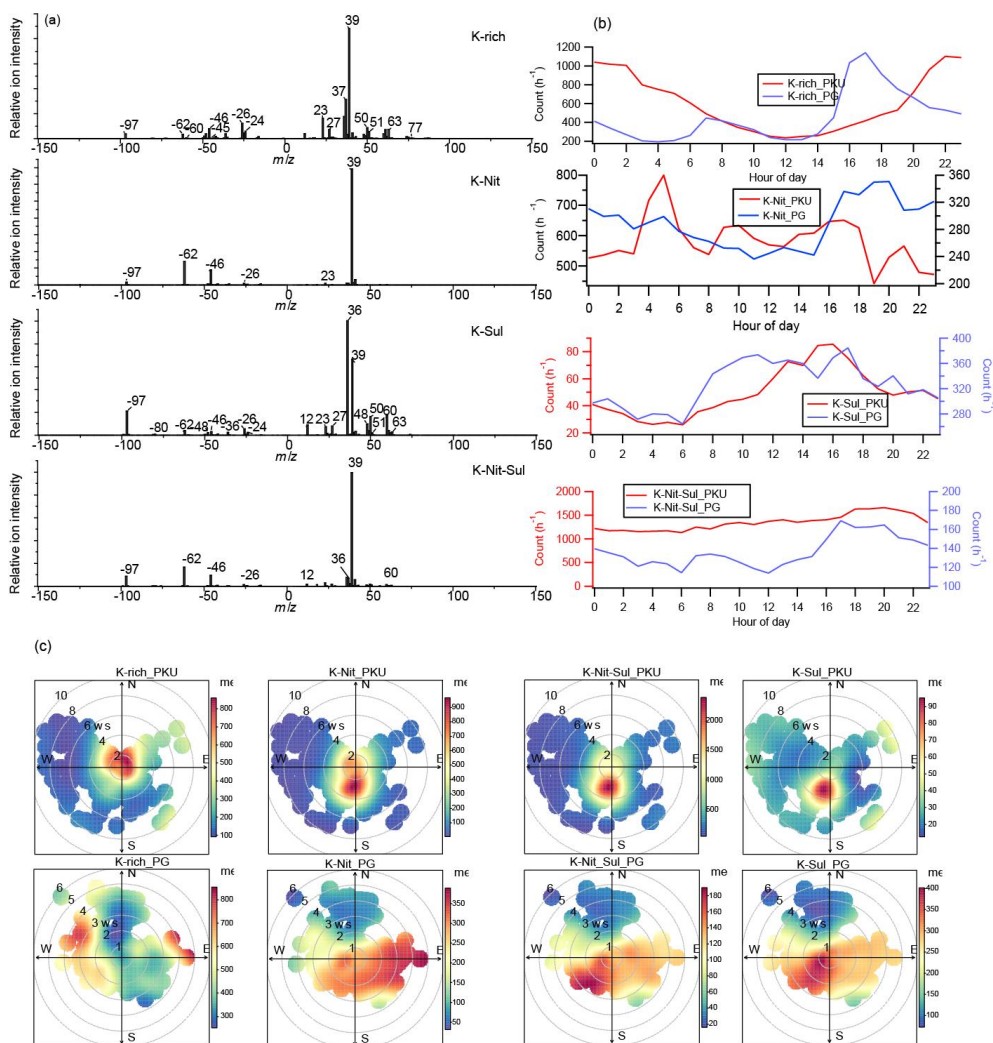

Figure 6. (a): average mass spectra of BB, K-Nit, K-Sul, and K-Nit-Sul observed at both

sites; (b): diurnal patterns of the hourly count of K-rich, K-Nit, K-Sul, and K-Nit-Sul at

both sites; (c): polar plots of BB, K-Nit, K-Sul, and K-Nit-Sul; the grey circles indicate

wind speed (m s$^{-1}$).





**3.2.5 Metal category**
Two metal-rich particles types were identified, namely Fe-rich and Ca-rich. Fe-rich
contained iron ($m/z$ 56 and 54), $K^+$, $Na^+$, $NH_4^+$, $Cl^-$ ($m/z$ -35 and -37), sulfate, and nitrate.
Ca-rich was composed of $Ca^+$ ($m/z$ 40), CaO ($m/z$ 56), $K^-$, $Na^+$, $Cl^-$, sulfate, and nitrate. As
shown in Figure 6b, Ca-rich_PKU (0.4%) and Ca-rich_PG (0.1%) were likely of regional
origin with no distinct diurnal variations. Since $SiO_2^-$ or $SiO_3^-$ ($m/z$ -60 and -76) were not
abundant in the Ca-rich particles, they are not likely to come from dust (Silva et al., 2000).
According to its weak peaks during the rush hour at PKU, a possible source of the Ca-rich
particles was from road dust re-suspension. Such rush hour peaks were not observed at PG.
Fe-rich_PKU (3.1%) and Fe-rich_PG (1.8%) had similar diurnal profiles that arose in the
early morning when heavy-duty vehicles were allowed to enter the 5-ring expressway. The
peak occurred earlier at PG (4:00) than (5:00) because these vehicles got close to PG earlier
than to PKU. The daytime peak occurred in the afternoon at both PKU and PG when wind
speed was high. Therefore, there were also multiple sources for Fe-rich particles, including
re-suspended dust particles from traffic and fly ash from the steel industry. In Beijing,
daytime Fe-rich particles were reported and assigned to long-range transport and industrial
sources from Heibei Province (Figure 7c) (Li et al., 2014). The steel industry moved out
of Beijing more than a decade ago (Liu et al., 2016b). Currently, most of these steel
industries were located in the Heibei Province.

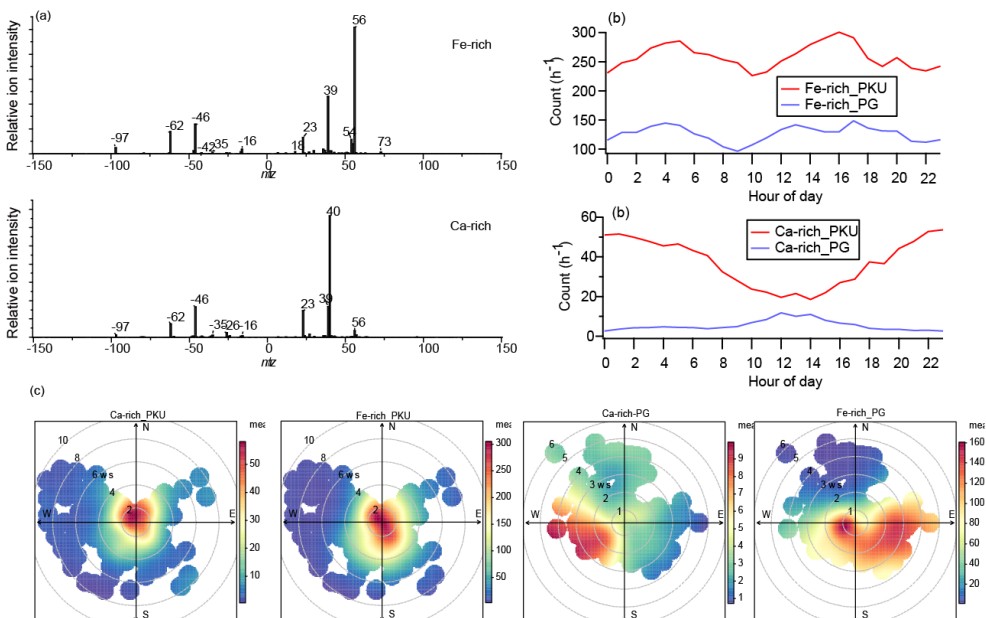


Figure 7. (a): average mass spectra of Fe-rich and Ca-rich observed at both sites; (b): diurnal patterns of the hourly count of Fe-rich and Ca-rich at both sites; (c): polar plots of Fe-rich and Ca-rich; the grey circles indicate wind speed ($ms^{-1}$).

### 3.2.6 NaK category

As shown in Figure 8, mass spectra of NaK category contained f $Na^+$, $K^+$, $C_n^+$, $C_n^-$, nitrate, and $PO_3^-$ (*m/z* -79). The aged NaK particles contained strong signals of nitrate (NaK-Nit), sulfate (NaK-Sul), or both (NaK-Nit-Sul). In general, the NaK category contained stronger signals of $Na^+$ than the EC and K-rich categories. The NaK category may also come from incomplete solid fuel combustion processes such as coal, peat, or wood (Chen et al., 2017; Healy et al., 2010; Xu et al., 2017). NaK category was more abundant at PKU (9.5%) than PG (5.8%), suggesting a stronger contribution of emission from coal boilers (Xu et al.,





2017; Xu et al., 2018). Additionally, after heating began, the fraction of NaK-Nit_PG and
NaK-Sul-Nit_PG increased by 1.2 times (see Part II).
NaK_PKU showed no distinct diurnal variations, suggesting that it was a regional particle
type arriving at the PKU site via transport, while NaK_PG showed an apparent diurnal
variation consistent with cooking and heating pattern. Polar plots also suggest that they are
from the east and the west. NaK-Nit, with a considerable uptake of nitrate, was more
abundant at PKU (6.4%) than PG (1.7%). Both NaK-Nit_PKU and NaK-Nit _PG increased
in the afternoon when photochemical activities were most active (Figure 8c). Both of them
may be from regional transport (Figures 8b and 8c).
NaK-Sul was a minor particle type at both PG and PKU, accounting for 0.2% and 0.4%,
respectively. The diurnal profile of NaK-Sul_PG was also following the local cooking and
heating pattern, while NaK-Sul_PKU showed a typical transport pattern that became
abundant in the afternoon as the southwestern wind speed increased. As a heavily aged
particle type, NaK-Nit-Sul was transported to both PKU and PG from the southwest. In
short, NaK-related particle types mainly came from the solid fuel burning process, e.g.,
coal. Due to its different origins, it showed different levels of processing at PKU and PG,
respectively.

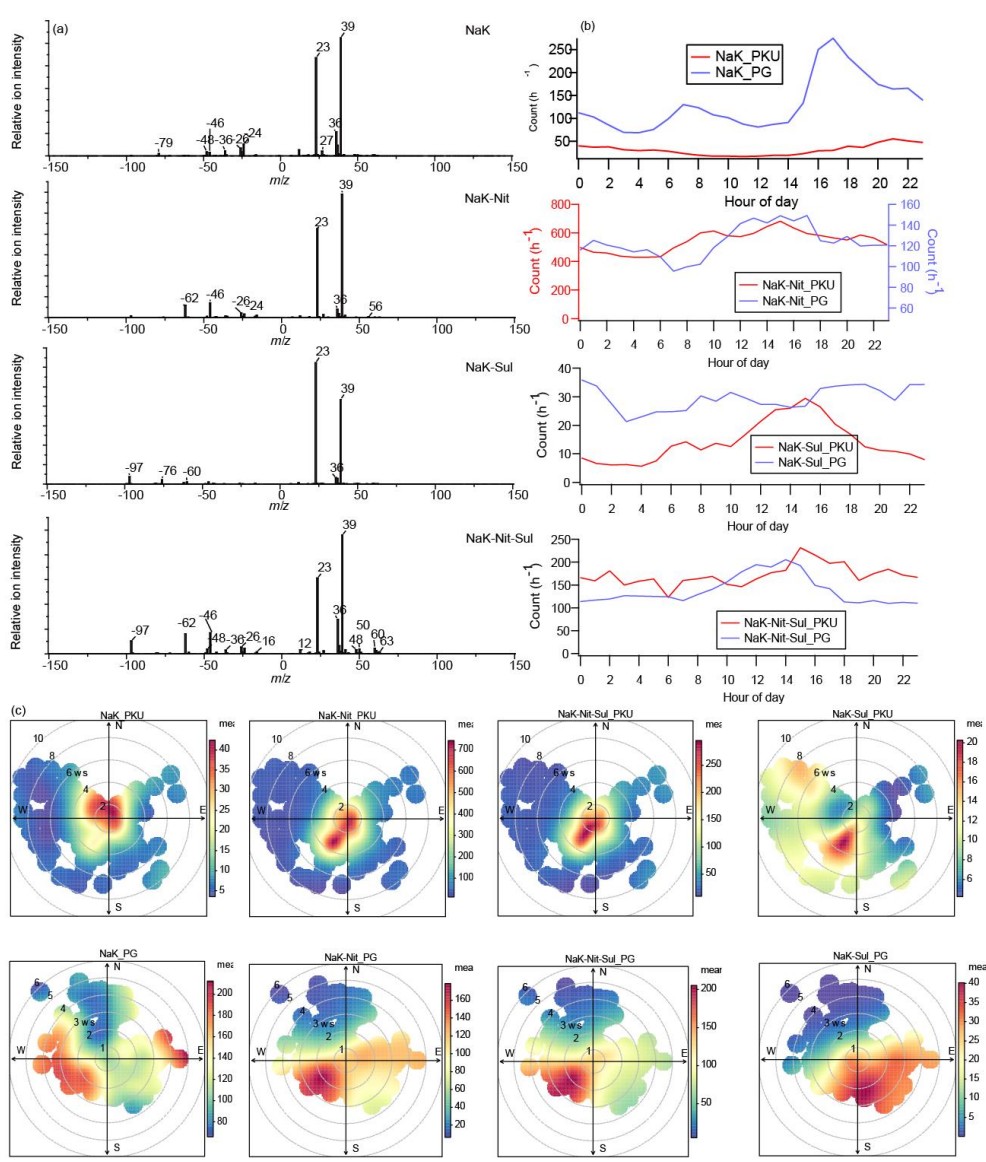


Figure 8. (a): average mass spectra of NaK, NaK-Nit, NaK-Nit-Sul, and NaK-Sul observed
at both sites; (b): diurnal patterns of the hourly count of NaK, NaK-Nit, NaK-Nit-Sul, and
NaK-Sul at both sites; (c): polar plots of NaK, NaK-Nit, NaK-Nit-Sul, and NaK-Sul; the
grey circles indicate wind speed (m s$^{-1}$).





**3.3 Unique Particle types at the PKU site**
OC-Nit_PKU (0.9%) and ECOC-Nit_PKU (3.1%) with strong ion intensities of nitrate
were observed at the PKU site. OC-Nit_PKU and ECOC-Nit_PKU showed a peak at night
than at daytime, similar to the diurnal profiles of OC-Nit-Sul_PKU and ECOC-Nit-
Sul_PKU. Such nitrate-rich particle types could have come from the uptake of nitrate in
OC and ECOC(Qin et al., 2012; Chen et al., 2016). Polar plots suggest that both types were
formed locally when the wind speed was lower than 4 ms$^{-1}$. The NO$_x$-rich environment in
urban Beijing provides a favorable condition for nitrate formation at night (Wang et al.,
2016; Zou et al., 2015; Shi et al., 2019).
A minor amount (0.10%) of amine-containing particles was observed at the PKU site, and
trimethylamine ion fragments (*m/z* 58 and 59) were influential in the mass spectrum of K-
amine-Nit-Sul_PKU (Figure 9a). The diurnal profile of K-amine-Nit-Sul_PKU showed an
afternoon peak, indicating a regional source (Figure 9c). K-amine-Nit-Sul_PKU was
transported to the site from nearby locations. The amines may come from animal husbandry,
BB, traffic, or vegetation (Chen et al., 2019). Amines were ubiquitous in the atmospheric
environment, playing essential roles in new particle formation and growth, as well as fog
and cloud processing (Ge et al., 2011; Chen et al., 2019).

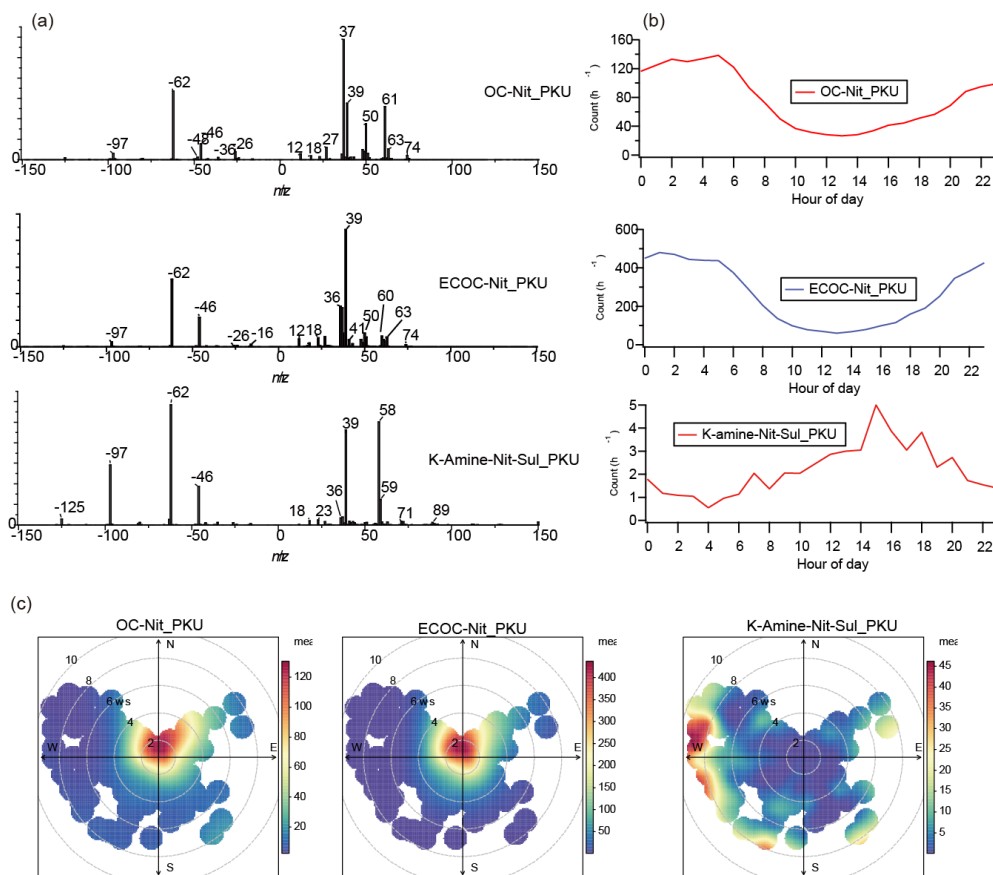


Figure 9. (a): average mass spectra of OC-Nit_PKU, ECOC-Nit_PKU, and K-amine-Nit-
Sul_PKU observed at the PKU site; (b): diurnal patterns of the hourly count of OC-
Nit_PKU, ECOC-Nit_PKU, and K-amine-Nit-Sul_PKU at the PKU site; (c): polar plots of
OC-Nit_PKU, ECOC-Nit_PKU, and K-amine-Nit-Sul_PKU, and the grey circles indicate
wind speed (m s$^{-1}$).

**3.4 Unique Particle types at the PG site**

OC_PG (5.9%) and ECOC_PG (3.3%) were only observed at the rural site PG (Figure 10).

The major components of these two types were consistent with the OC and ECOC





categories, respectively, but with limited uptake of sulfate and nitrate, suggesting that they
were possibly freshly emitted particles. Their diurnal profiles are consistent with cooking
and heating patterns which peaked at 07:00 in the morning and 17:00. Polar plots suggest
that OC_PG mainly came from nearby and other remote areas in all directions except the
north. ECOC mainly came from the east of the PG site. These results supported the
assumption that the two types were mainly from local emission sources.

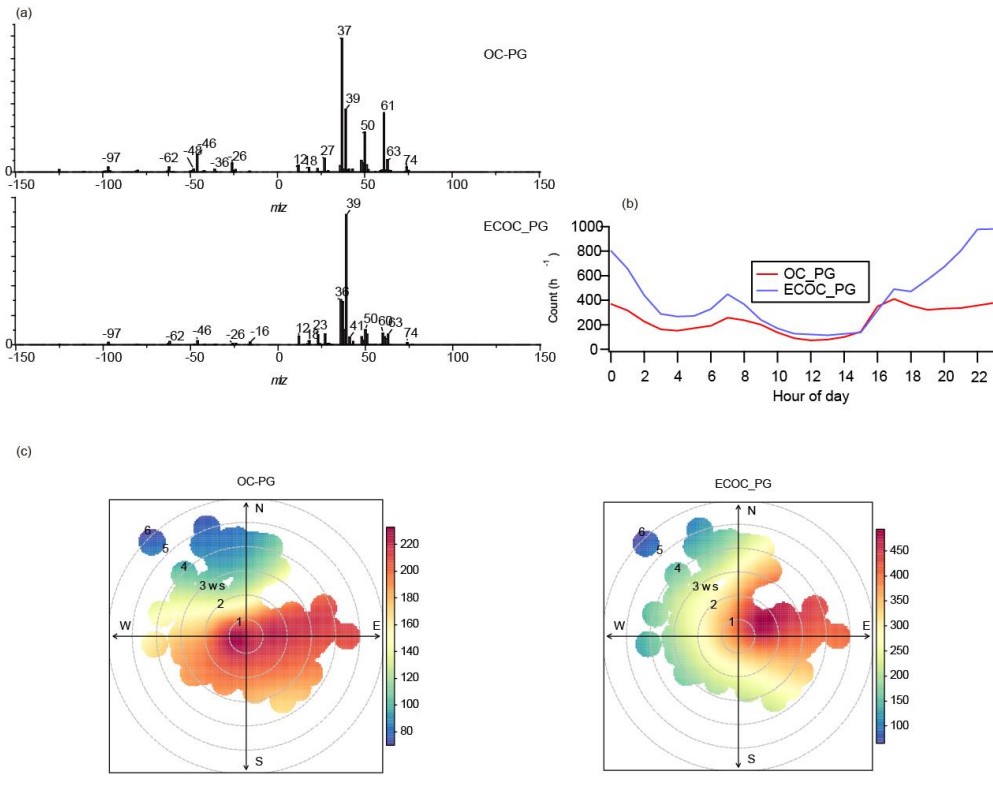

Figure 10. Average mass spectra of (a) ECOC-Nit, (b) ECOC-Nit, and (c) OC-Nit. All
these particle types appeared at the PG site.





## 4. Discussion

Multiple source apportionment models have been used in Beijing to quantify the sources of particles (Sun et al., 2014; Xu et al., 2015; Zhai et al., 2016). Biomass burning, coal combustion, traffic, and dust are the key sources of PM (Sun et al., 2014; Liu et al., 2018; Huang et al., 2014). Multiple studies confirmed that biomass burning is an essential source of PM in urban Beijing (Gao et al., 2014; Huang et al., 2014; Sun et al., 2014; Zheng et al., 2017). In this study, biomass burning, and other solid fuel burning were identified as crucial sources of PM in not only urban but also rural areas of Beijing. We observed that BB-related particles (K-rich category) were more abundant at PG than at PKU. In particular, we found fresh-emitted K-containing particles at the Pinggu site, confirmed the importance of local emissions to PM. Furthermore, K-containing particles in the urban area were more aged, suggested that they are aged and mostly from the surrounding areas. The result is consistent with the results from (Liu et al., 2019) based on a combined receptor and footprint models. Nevertheless, household emissions in the BHT region caused 32% and 15% of primary $PM_{2.5}$ and $SO_2$. These studies have proved the importance of household emission from BB in the BHT area (Liu et al., 2016a). Especially at the PG site, the ambient PM was mainly controlled by long-range transport and household emissions from cooking and heating.

Due to the nature of SPAMS, the chemical composition of PM cannot be precisely quantified. However, single particle aerosol mass spectrometers have advantages in studying the chemical composition, mixing state, source, and process of particles (Pratt and Prather, 2012). Mass-based technologies can not differentiate the origin of the bulk of nitrate, whether it is transported or formed locally. Indeed, single particle types in urban





Beijing have been reported in previous studies (Li et al., 2014; Liu et al., 2016b), and the
major types are consistent with this study. However, in this study, we adopted a cluster
strategy considering the relative ion peak area of sulfate and nitrate as indicators of particle
processing. Therefore, more detailed could be extracted from both two simultaneous
datasets. We confirmed that the source, origination, and processes were different for these
particles in the urban and rural areas. For example, the seriously processed particles, such
as K-Nit-Sul, OC-Nit-Sul, and NaK-Nit-Sul, acted with no distinct diurnal patterns as
background or regional sources. The processed particles, such as OC-Nit, ECOC-Nit, and
NaK-Nit, were affected by emissions and secondary formations.
The emission and transport patterns were different in the urban and rural areas, resulting in
different characteristics of PM. For example, EC particles were a key component at PKU
(18.2% in total), but a minor particle type at PG (5.6%). Meanwhile, in the urban area of
Beijing, direct emission of K-rich particles should be small due to strict control measures;
thus, the K-Nit-Sul particles are mainly from long-range transport. More interestingly, the
sulfate-rich particle types, such as EC-Sul and ECOC-Sul, were commonly accompanied
by high wind speed, suggesting the contribution of regional air masses. Sulfate and $SO_2$
emissions were controlled strictly in Beijing. However, in the nearby Hebei and Shandong
province, the emission of $SO_2$ is still significant (Shi et al. 2019). Transported particles
were aged and commonly coated a thick layer of nitrate and sulfate, but the local particles
were affected by both emission and the near-surface aging process. For example, at PKU,
the primary emission sources were traffic and central heating supply, causing a $NO_x$-rich
region in which freshly-emitted particle types could undergo processing due to the uptake
of nitrate (Wang et al., 2016). In the nearby villages of PG, domestic heating and cooking





were the major contributors of primary particles when the temperature was low in the
morning and afternoon, resulting in the emission of multiple primary particle types such as
OC_PG and ECOC_PG. In short, the characteristics of PM in urban and rural areas of
Beijing were affected by local emissions and interacted with each other due to regional
transport.
Secondary nitrate formation is still a critical issue in Beijing. The daytime arising of nitrate
has been reported in studies (Sun et al., 2013), and we also found a similar predominant of
nitrogen-containing particles in this study. Recent studies have reported the early morning
peaks of nitrate using a soot particle aerosol mass spectrometer (SP-AMS) (Wang et al.,
2019), which is consistent with our results. Interestingly, the early morning peak was only
observed for several particle types at both sites, including EC-Nit_PKU, K-Nit_PKU, EC-
Nit-Sul-PG, and EC-Nit_PG. This result is not surprising because PG is also a $NO_2$-rich
region (Shi et al., 2019). The increasing contribution of nitrate-containing particles
suggests the role of night chemistry in nitrate uptake on particles. Wang et al. (2017)
revealed the importance of night $N_2O_5$ chemistry on nocturnal nitrate formation in the
urban area of Beijing. The heterogeneous hydrolysis of $N_2O_5$ was most favorable when NO
was at a low level. Moreover, the polar plots suggested a small role of long-range transport
to the nitrate in individual particles. The contribution of local traffic was insignificant at
the PG site as it was far from highways and major roads, the nighttime formation of nitrate
appeared to be important in PG as well.



## 5. Conclusion

Two SPAMSs were simultaneously deployed at urban and rural sites in Beijing in order to characterize PM during wintertime. The results at both sites indicate that they shared 17 types of common clusters, most of which belonged to particle categories such as EC, OC, ECOC, BB, and NaK. The origins and sources of these particle types at both sampling sites are also comprehensively analyzed. Most of the processed PM, including EC-Nit-Sul_PKU, ECOC-Nit-Sul_PKU, and NaK-Nit-Sul_PKU, were aged locally in a $NO_x$-rich environment, while EC-Nit-Sul_PG, ECOC-Nit-Sul_PG, NaK-Nit-Sul_PG, and OC-Nit-Sul_PG were regional. Domestic heating in the rural area was found to be an important source of PM, and such heating activities typically caused three diurnal peaks in the early morning, morning, and afternoon (after sunset). Moreover, the early morning peak of nitrate was observed at both sites, suggesting the contribution of the heterogeneous hydrolysis of $N_2O_5$ in the dark during the winter. The insights gained in this study can provide useful references for understanding the relationship between regional transport and local aging in both urban and rural areas in Beijing. In Part II, we focus on haze events observed at both sites and attempt to determine the effects of heating activities and possible regional transport between PKU and PG.

*Data availability.* All data described in this study are available upon request from the corresponding authors.

*Author contributions.* FY, MZ, TZ, QZ, and KH designed the experiments; YC, JC, ZW, MT, CP, and HY carried them out; XYang, XYao, YL, GS, and ZS analyzed the experimental data; YC prepared the manuscript with contributions from all coauthors.



*Competing interests.* The authors declare that they have no conflict of interest.
*Acknowledgments.* We are grateful for financial support from the National Natural Science
Foundation of China (Grant No. 4170030287 and 81571130100). ZS acknowledges
funding from NERC (NE/N007190/1 and NE/R005281/1).

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
