# Peer review of "Simultaneous Measurement of Urban and Rural Single Particles in Beijing, Part I: Chemical Composition and Mixing State"

_Atmospheric Chemistry and Physics, 2019_

## Referee Comment (RC1) · Anonymous Referee #3 · 21 Feb 2020

This work did a nice job in measuring the chemical composition and mixing states of aerosols at an urban and a rural site in Beijing. More than 4 million particles were detected at each site, of which the chemically analyzed particles were grouped and analyzed to investigate the potential sources and atmospheric processing. The authors found that the urban particles were influenced significantly by rural processing and transport. The paper is generally well-written, and I recommend it for publication after some addressing the following comments.

Comments: 1. The authors listed several studies in paragraph 2 in Introduction, but didn't contain any conclusion. What does it mean by "discrepancies remain among

these studies"? Authors need to be more specific about what discrepancies exist among the studies. Please revise/rephrase, and be more specific.

2. It should be noted that chemical bias of SPAMS might introduce uncertainties in representativeness of ambient particles and even in classification of chemically analyzed particles. I think there is a need to mention this caveat in your paper, and caution the readers that uncertainties may be expected for the results. This can be provided either in Introduction or Discussions.

3. Please clarify the differences between this study and previous studies. The significance of this study is not well written in the paper.

4. L150: The EC category has four types including EC-Nitrate (EC-Nit), EC-Sulfate (EC-Sul), and EC-Nit-Sul. Is it "four", not "three"? Or there is another type?

5. According to line 243, K-rich is one of branches of k-rich category. According to Figure 6a, K-rich means BB (BB in figure 6 caption, and K-rich in the figure) According to line 366, BB-related particle means K-rich category. The use of K-rich category, K-rich type, BB and BB-related particle is confusing. Please make the description more clear, specific and concrete.

6. As you mentioned in Line 253, the household BB is prohibited in urban Beijing, which is inconsistent with figure 6c indicating that the highest number counts of K-rich were observed when wind speed was less than 2 m s-1, is that possible K-rich_PKU can also from other sources?

7. L295: "mass spectra of NaK category contained f Na+, K+ ..." Please correct.

---

## Referee Comment (RC2) · Anonymous Referee #4 · 14 May 2020

The authors performed simultaneous observations of size-resolved single-particle chemical composition and mixing state in both urban and rural areas of Beijing. The campaigns collected more than 4 million particles being at both sites. The authors have investigated the different sources, processing, and origination of atmospheric particles at both sites. Generally, the manuscript illustrates a substantial contribution to the scientific understanding of urban particulate pollution in China. Particularly, the clustering strategy used in this work can be very useful to illustrate different stages of atmospheric processing. The manuscript is concisely organized and well written. Therefore, the reviewer prefers publishing if the following concerns are addressed.

[Figure]

Major comments

1. The statement in the Introduction should be more clarified between Lines 47‒53. The discrepancies should be described, and a conclusion or hypothesis should have been drawn from the literature review. 2. Also, could you be specific on this "limited attention to the atmospheric particulate processing?" It is very important because it provides the novelty of this work compared to previous studies. 3. Lines 101‒112. The use of relative peak area to determine the aging of particles is interesting. As we know, matrix effect could alter the ion intensities of each ion. The authors should address the possible influential factor for bias. 4. Table 1. The reviewer strongly suggests a column of comment on the source of particle types at both sites, respectively. 5. Section 3.2.5. Is it possible to use the polar plots at both sites to locate the potential source of Fe-rich particles? 6. Please compare the -Sul particles at both sites because the sulfate-rich particles can be formed from the oxidation of SO2. Any difference in the origin of sulfate-rich particles at different sites? Please add additional analysis or comments to Discussion. 7. Section 3.4. according to the claim in the manuscript, both OC-PG and ECOC-PG are supposed to be local. However, the polar plot suggests that these particle types are from multiple directions, please explain.

Minor and technical

Lines 385: "detailed" should be "details." Line 421. "small" should be "limited."
* * *

---

## Author Comment (AC1) · 23 May 2020

Dear reviewer,

The authors are thankful for the reviewer's comments, suggestions, and time. We have prepared a point-by-point response highlighted in blue. We hope our efforts in revising the manuscript can improve it for the selection of ACP.

Please also see the Supplement for a pdf file.

Anonymous Referee #3

This work did a nice job in measuring the chemical composition and mixing states of

aerosols at an urban and a rural site in Beijing. More than 4 million particles were detected at each site, of which the chemically analyzed particles were grouped and analyzed to investigate the potential sources and atmospheric processing. The authors found that the urban particles were influenced significantly by rural processing and transport. The paper is generally well-written, and I recommend it for publication after some addressing the following comments.

Comments: 1. The authors listed several studies in paragraph 2 in Introduction, but didn't contain any conclusion. What does it mean by "discrepancies remain among these studies"? Authors need to be more specific about what discrepancies exist among the studies. Please revise/rephrase and be more specific.

Ans: We have added the following passage to illustrate the discrepancies and gaps in knowledge (lines 53-62): "For example, the mass loading of PM2.5 can rapidly increase to hundreds $\mu$g m-3. Both Wang et al. (2016) and Cheng et al. (2016) suggested the secondary formation of sulfate from the oxidation of NO2, while Guo et al. (2014) proposed a mechanism of particle formation and growth. Different from local secondary formation and accumulation, Li et al. (2015) proposed that particles via long-range transport cause the elevation of PM2.5. According to Sun et al. (2014) and Zhai et al. (2016), regional transport played an important role during heavy haze episodes. However, most studies have focused on the urban areas of Beijing, with limited attention paid to rural areas. To illustrate the sources, evolution, and transport of particles, the investigation of rural areas around Beijing is necessary."

2. It should be noted that chemical bias of SPAMS might introduce uncertainties in representativeness of ambient particles and even in classification of chemically analyzed particles. I think there is a need to mention this caveat in your paper, and caution the readers that uncertainties may be expected for the results. This can be provided either in Introduction or Discussions.

Ans: We completely agree with the reviewer that the limitations of the instrumentation

should be addressed (lines70-72). "Due to the nature of laser desorption/ionization (LDI), the instrument is very sensitive to dust and other types of particles containing sodium and potassium, and this may cause bias in the particle matrix (Pratt and Prather, 2012)."

3. Please clarify the differences between this study and previous studies. The significance of this study is not well written in the paper.

Ans: Associated with what we mentioned in Lines 76-83, we have added the following paragraph: "Organics, sulfate, nitrate, ammonium and other species have been found internally mixed in the atmospheric particles, and these particle types are mostly from the combustion of fuel or biomass. The abundance of secondary species can indicate the degree of aging during atmospheric processing. Particles are more secondary species with deeper processing. However, these studies lack the use of this data to provide a view of the dynamic particulate processing. Therefore, we used the relative abundance of secondary species to adequately illustrate the process of single particles at both sites, providing a tracing system on a regional scale."

4. L150: The EC category has four types including EC-Nitrate (EC-Nit), EC-Sulfate (EC-Sul), and EC-Nit-Sul. Is it "four", not "three"? Or there is another type?

Ans: Sorry, that was is a typo. We have changed it to "three."

5. According to line 243, K-rich is one of branches of k-rich category. According to Figure 6a, K-rich means BB (BB in figure 6 caption, and K-rich in the figure) According to line 366, BB-related particle means K-rich category. The use of K-rich category, K-rich type, BB and BB-related particle is confusing. Please make the description more clear, specific and concrete.

Ans: Thank you very much for this suggestion. The K-rich category is from BB-related particles, and we will stick to this term. We have changed our description in the text, as well as the caption of Figure 6a.

6. As you mentioned in Line 253, the household BB is prohibited in urban Beijing, which is inconsistent with figure 6c indicating that the highest number counts of K-rich were observed when wind speed was less than 2 m s-1, is that possible K-rich_PKU can also from other sources?

Ans: This is a very interesting question.

The urban sampling site is around the 4th ring expressway of Beijing, 10 km from the 5th ring expressway. In these areas, there are still villages in which biofuels are used. Therefore, these K-rich_PKU were also from biomass burning.

7. L295: "mass spectra of NaK category contained f Na+, K+ . . ." Please correct. Ans: We have made the correction (line 319).

References

Cheng, Y., Zheng, G., Wei, C., Mu, Q., Zheng, B., Wang, Z., Gao, M., Zhang, Q., He, K., and Carmichael, G.: Reactive nitrogen chemistry in aerosol water as a source of sulfate during haze events in China, Science Advances, 2, e1601530, 2016.

Guo, S., Hu, M., Zamora, M. L., Peng, J., Shang, D., Zheng, J., Du, Z., Wu, Z., Shao, M., Zeng, L., Molina, M. J., and Zhang, R.: Elucidating severe urban haze formation in China, Proc Natl Acad Sci U S A, 111, 17373-17378, 10.1073/pnas.1419604111, 2014.

Li, P., Yan, R., Yu, S., Wang, S., Liu, W., and Bao, H.: Reinstate regional transport of PM2.5 as a major cause of severe haze in Beijing, Proc Natl Acad Sci U S A, 112, 2015.

Pratt, K. A., and Prather, K. A.: Mass spectrometry of atmospheric aerosols–recent developments and applications. Part II: On-line mass spectrometry techniques, Mass Spectrom. Rev., 31, 17-48, 10.1002/mas.20330, 2012.

Sun, Y., Jiang, Q., Wang, Z., Fu, P., Li, J., Yang, T., and Yin, Y.: Investigation of the

sources and evolution processes of severe haze pollution in Beijing in January 2013, J. Geophys. Res. Atmos., 119, 4380-4398, 10.1002/2014jd021641, 2014.

Wang, G., Zhang, R., Gomez, M. E., Yang, L., Zamora, M. L., Hu, M., Lin, Y., Peng, J., Guo, S., and Meng, J.: Persistent sulfate formation from London Fog to Chinese haze, Proceedings of the National Academy of Sciences, 113, 13630-13635, 2016.

Zhai, S., An, X., Zhao, T., Sun, Z., Hou, Q., and Wang, C.: Detecting critical PM2.5 emission sources and their contributions to a heavy haze episode in Beijing, China by using an adjoint model, Atmospheric Chemistry and Physics Discussions, 2016, 1-20, 10.5194/acp-2016-911, 2016.

Please also note the supplement to this comment:
https://www.atmos-chem-phys-discuss.net/acp-2019-933/acp-2019-933-AC1-supplement.pdf

---

## Author Comment (AC2) · 23 May 2020

Dear reviewer,

We are very grateful that you reviewed this manuscript. We appreciate your positive feedback on this work, and your comments and suggestions are very valuable for us in improving this study. We have prepared a detailed point-by-point response highlighted in blue. We hope our efforts in revising the manuscript can improve it for the selection of the journal.

Please also see the Supplement for a pdf file.

[Figure]

The authors performed simultaneous observations of size-resolved single-particle chemical composition and mixing state in both urban and rural areas of Beijing. The campaigns collected more than 4 million particles being at both sites. The authors have investigated the different sources, processing, and origination of atmospheric particles at both sites. Generally, the manuscript illustrates a substantial contribution to the scientific understanding of urban particulate pollution in China. Particularly, the clustering strategy used in this work can be very useful to illustrate different stages of atmospheric processing. The manuscript is concisely organized and well written. Therefore, the reviewer prefers publishing if the following concerns are addressed.

Major comments

1. The statement in the Introduction should be more clarified between Lines 47‑53. The discrepancies should be described, and a conclusion or hypothesis should have been drawn from the literature review.

Ans: The following passage has been added to the text (lines 53‑62): "For example, the mass loading of PM2.5 can rapidly increase to hundreds $\mu$g m‑3. Both Wang et al. (2016) and Cheng et al. (2016) suggested the secondary formation of sulfate from the oxidation of NO2, while (Guo et al., 2014) proposed a mechanism of particle formation and growth. Different from local secondary formation and accumulation, Li et al. (2015) proposed that particles via long-range transport cause the elevation of PM2.5. According to Sun et al. (2014) and Zhai et al. (2016), regional transport plays an important role during heavy haze episodes. However, most studies have focused on the urban areas of Beijing, with limited attention paid to rural areas. To illustrate the sources, evolution, and transport of particles, the investigation of rural areas around Beijing is necessary."

2. Also, could you be specific on this "limited attention to the atmospheric particulate processing?" It is very important because it provides the novelty of this work compared to previous studies.

Ans: We completely agree with the reviewer. The following statement has been added to the text to enhance the argument (lines 76‒83): "Organics, sulfate, nitrate, ammonium and other species have been found internally mixed in the atmospheric particles, and these particle types are mostly from the combustion of fuel or biomass. The abundance of secondary species can indicate the degree of aging during atmospheric processing. Particles are with more secondary species with deeper processing. However, these studies lack the use of this data to provide a view of the dynamic particulate processing. Therefore, we used the relative abundance of secondary species to adequately illustrate the process of single particles at both sites, providing a tracing system on a regional scale."

3. Lines 101‒112. The use of relative peak area to determine the aging of particles is interesting. As we know, matrix effect could alter the ion intensities of each ion. The authors should address the possible influential factor for bias.

Ans: We have added this part to the text (lines 128‒131): "Indeed, the matrix effect can affect ionic intensities between different particles during single-particle mass spectrometer analysis. However, the effect can be reduced by using the average mass spectra of particles within a similar size distribution and chemical composition."

4. Table 1. The reviewer strongly suggests a column of comment on the source of particle types at both sites, respectively.

Ans: We have added a comment on the sources of particles.

5. Section 3.2.5. Is it possible to use the polar plots at both sites to locate the potential source of Fe-rich particles?

Ans: Yes, according to the polar plots at both sites, the Fe-rich particles originated from the south of both sites which is the direction of Hebei Province, as we described in the text.

6. Please compare the -Sul particles at both sites because the sulfate-rich particles

can be formed from the oxidation of SO2. Any difference in the origin of sulfate-rich particles at different sites? Please add additional analysis or comments to Discussion.

Ans: A detailed interpretation of sulfate-rich particles has been added in lines 433-446: "SO2 was strictly controlled in Beijing. However, the emission of SO2 is still significant in the nearby Hebei and Shandong provinces (Shi et al. 2019). The different control measures produced a low concentration area of SO2 around Beijing. Sulfate-rich particle types such as EC-Sul, OC-Sul, K-Sul, and NaK-Sul usually arrived at the PKU site when the wind speed was high (> 3m s‒1). The wind directions, along with the transport of sulfate-rich particles, were east, southwest and south. In these directions, sulfate was either primarily emitted from coal burning for residential heating, power generation and industry, or secondary uptake on pre-existing particles (Zhang et al., 2015). Likewise, a portion of sulfate-rich particle arrived at the PG site when the wind speed was high. However, locally formed sulfate was also pronounced, especially for ECOC-Sul, K-Sul, and NaK-Sul. As discussed in Section 3, ECOC-Sul and NaK-Sul were mainly from coal burning for residential heating, and K-Sul was formed due to the uptake of secondary sulfate. Conclusively, the particulate characterization in rural areas around Beijing is significantly influenced by residential coal burning."

7. Section 3.4. according to the claim in the manuscript, both OC-PG and ECOC-PG are supposed to be local. However, the polar plot suggests that these particle types are from multiple directions, please explain.

Ans: Yes, OC_PG was mainly from the east, south, and west, and ECOC_PG came from the northeast, southeast, and east. They certainly came from multiple directions. However, the highest concentrations of these particles were at the centers of the polar plots, indicating that emissions of OC_PG and ECOC_PG were high in the Beijing region.

Therefore, the following statement has been added to line 381: "Also, the emissions of OC_PG and ECOC_PG are high in the region."

Minor and technical

Lines 385: "detailed" should be "details."

Ans: We have changed this part (line 412).

Line 421. "small" should be "limited."

Ans: We have changed this part (line 421).

References

Cheng, Y., Zheng, G., Wei, C., Mu, Q., Zheng, B., Wang, Z., Gao, M., Zhang, Q., He, K., and Carmichael, G.: Reactive nitrogen chemistry in aerosol water as a source of sulfate during haze events in China, Science Advances, 2, e1601530, 2016.

Guo, S., Hu, M., Zamora, M. L., Peng, J., Shang, D., Zheng, J., Du, Z., Wu, Z., Shao, M., Zeng, L., Molina, M. J., and Zhang, R.: Elucidating severe urban haze formation in China, Proc Natl Acad Sci U S A, 111, 17373-17378, 10.1073/pnas.1419604111, 2014.

Li, P., Yan, R., Yu, S., Wang, S., Liu, W., and Bao, H.: Reinstate regional transport of PM2.5 as a major cause of severe haze in Beijing, Proc Natl Acad Sci U S A, 112, 2015.

Sun, Y., Jiang, Q., Wang, Z., Fu, P., Li, J., Yang, T., and Yin, Y.: Investigation of the sources and evolution processes of severe haze pollution in Beijing in January 2013, J. Geophys. Res. Atmos., 119, 4380-4398, 10.1002/2014jd021641, 2014.

Wang, G., Zhang, R., Gomez, M. E., Yang, L., Zamora, M. L., Hu, M., Lin, Y., Peng, J., Guo, S., and Meng, J.: Persistent sulfate formation from London Fog to Chinese haze, Proceedings of the National Academy of Sciences, 113, 13630-13635, 2016.

Zhai, S., An, X., Zhao, T., Sun, Z., Hou, Q., and Wang, C.: Detecting critical PM2.5 emission sources and their contributions to a heavy haze episode in Beijing, China by

using an adjoint model, Atmospheric Chemistry and Physics Discussions, 2016, 1-20, 10.5194/acp-2016-911, 2016.

Zhang, R., Wang, G., Guo, S., Zamora, M. L., Ying, Q., Lin, Y., Wang, W., Hu, M., and Wang, Y.: Formation of urban fine particulate matter, Chem. Rev., 115, 3803-3855, 10.1021/acs.chemrev.5b00067, 2015.

Please also note the supplement to this comment:
https://www.atmos-chem-phys-discuss.net/acp-2019-933/acp-2019-933-AC2-supplement.pdf

---

## Author Response (AR2)

Dear Prof. Frank Keutsch,

We are grateful for the technical corrections. We have prepared a point-by-point response to the detailed corrections, and they have been changed accordingly.

Comments to the Author:

Dear authors,

thank you for your detailed responses to the reviewers' comments and revisions of the manuscript. I just have a few additional technical corrections:

P. 4 line 72 "and this may cause bias in the particle matrix" could you make a little clearer what you mean with "bias in the particle matrix". Do you mean matrix effects can bias relative intensities? Please specify.

Yes, and we have changed it to " "

"…this may cause bias in the matrix of ionic intensities of chemical species."

P. 4 line 79 "Particles are with more secondary species with deeper processing."

. Consider changing to "Particles with higher amounts of secondary species have been processed more extensively" Also, does it matter what secondary species. I am not sure that more sulfate indicates more aging/processing.

We have changed the part.

Sulfate is a typical secondary species in China (Tao et al., 2017) . Uptake of secondary sulfate from the oxidation of SO2 via multiple pathways has caused severe haze in China. The relationship between aging and sulfate uptake has also been discussed in the literature (Li et al., 2011).

P. 4 line 80 "However, these studies lack the use of this data to provide a view of the dynamic particulate processing." Please be more specific on "these studies" and "this data".

we have fixed the part:

However, the previous studies mentioned above lack the use of data of relative intensities of secondary species to provide a view of the dynamic particulate processing.

P. 4 line 82 "the process of single particles" Do you mean the "processing" if not I don't understand what "the process of single particles"p. means.

Yes, it is processing, we have fixed the misuse.

P. 26 line 381 "Also, the emission of OC_PG and ECOC_PG is popular in the region." Popular implies that people like doing this. I would suggest using "common" or "frequent" instead.

We have changed it to "common."

P. 29 line 435 "a low concentration area of SO2". I suggest rephrasing as "an area of low SO2 concentration ..."

We have fixed it.

P. 29 lines 445-446 "Conclusively, the particulate characterization in rural areas around Beijing is significantly influenced by residential coal burning." I would suggest using "composition" rather than "characterization". I think "characterization" is the act of analyzing it and I think you mean that it is the actual "composition" or something similar that is influenced.

Thank you very much. We have changed it to "composition."

Please do not hesitate to contact me if you have any questions.

Best Regards,

Frank Keutsch

References

Li, W., Zhou, S., Wang, X., Xu, Z., Yuan, C., Yu, Y., Zhang, Q., and Wang, W.: Integrated evaluation of aerosols from regional brown hazes over northern China in winter:

Concentrations, sources, transformation, and mixing states, J. Geophys. Res., 116, 10.1029/2010jd015099, 2011.

Tao, J., Zhang, L., Cao, J., and Zhang, R.: A review of current knowledge concerning PM2. 5 chemical composition, aerosol optical properties and their relationships across China, Atmos. Chem. Phys., 17, 9485-9518, 10.5194/acp-17-9485-2017, 2017.